# Green Synthesis of Low-Glycemic Amylose–Lipid Nanocomposites by High-Speed Homogenization and Formulation into Hydrogel

**DOI:** 10.3390/molecules28207154

**Published:** 2023-10-18

**Authors:** Nasir Mehmood Khan, Misbah Uddin, Ebenezer Ola Falade, Farman Ali Khan, Jian Wang, Muhammad Shafique, Reem M. Alnemari, Maram H. Abduljabbar, Shujaat Ahmad

**Affiliations:** 1Department of Agriculture, Shaheed Benazir Bhutto University, Sheringal, Upper Dir 18000, Khyber Pakhtunkhwa, Pakistan; nasir@sbbu.edu.pk; 2Department of Chemistry, Shaheed Benazir Bhutto University, Sheringal, Upper Dir 18000, Khyber Pakhtunkhwa, Pakistan; roghanidoag136@gmail.com (M.U.); farmanali@sbbu.edu.pk (F.A.K.); 3Institute of Food Science and Technology, Chinese Academy of Agriculture Sciences, Beijing 100193, China; faladeebenezerola@gmail.com; 4Department of Chinese Materia Medica, Chongqing College of Traditional Chinese Medicine, Chongqing 402760, China; wj_2000_abc@cqmu.edu.cn; 5College of Traditional Chinese Medicine, Chongqing Medical University, Chongqing 400016, China; 6Department of Pharmaceutical Sciences, College of Pharmacy, Shaqra University, Shaqra 15571, Saudi Arabia; 7Department of Pharmaceutics and Pharmaceutical Technology, College of Pharmacy, Taif University, P.O. Box 11099, Taif 21944, Saudi Arabia; r.alnemari@tu.edu.sa; 8Department of Pharmacology and Toxicology, College of Pharmacy, Taif University, P.O. Box 11099, Taif 21944, Saudi Arabia; maram.a@tu.edu.sa; 9Department of Pharmacy, Shaheed Benazir Bhutto University, Sheringal, Upper Dir 18000, Khyber Pakhtunkhwa, Pakistan

**Keywords:** amylose–lipid nanocomposite, hydrogel, rheology, glycemic index (GI), absorption capacity

## Abstract

In this research, we focused on the production of amylose–lipid nanocomposite material (ALN) through a green synthesis technique utilizing high-speed homogenization. Our aim was to investigate this novel material’s distinctive physicochemical features and its potential applications as a low-glycemic gelling and functional food ingredient. The study begins with the formulation of the amylose–lipid nanomaterial from starch and fatty acid complexes, including stearic, palmitic, and lauric acids. Structural analysis reveals the presence of ester carbonyl functionalities, solid matrix structures, partial crystallinities, and remarkable thermal stability within the ALN. Notably, the ALN exhibits a significantly low glycemic index (GI, 40%) and elevated resistance starch (RS) values. The research extends to the formulation of ALN into nanocomposite hydrogels, enabling the evaluation of its anthocyanin absorption capacity. This analysis provides valuable insights into the rheological properties and viscoelastic behavior of the resulting hydrogels. Furthermore, the study investigates anthocyanin encapsulation and retention by ALN-based hydrogels, with a particular focus on the influence of pH and physical cross-link networks on the uptake capacity presenting stearic-acid (SA) hydrogel with the best absorption capacity. In conclusion, the green-synthesized (ALN) shows remarkable functional and structural properties. The produced ALN-based hydrogels are promising materials for a variety of applications, such as medicine administration, food packaging, and other industrial purposes.

## 1. Introduction

Starch is a biopolymer that is widely distributed in nature and is considered a major storage carbohydrate synthesized and accumulated in photosynthetic organisms. It is a versatile material with remarkable features, such as biocompatibility, biodegradability, low toxicity, low cost, and widespread availability, making it suitable for various applications, including food, pharmaceuticals, biomedical, and many others [1]. However, cereals and other starchy foods, which are commonly consumed as refined flour, have a high glycemic index and a low amount of dietary fiber. This can lead to diet-related disorders such as type 2 diabetes, obesity, cardiovascular disease, and cancer [2]. As a result, numerous government agencies have emphasized the necessity for refined low-glycemic food production and the benefits of consuming more whole grains and pulses while reducing the intake of refined starchy foods [3]. Various techniques, including chemical, enzymatic, and thermomechanical methods, have been used to change the physicochemical and functional characteristics of starches. Previously, amylose–lipid complexes have been prepared by various techniques, including heating in ethanol/NaOH, DMSO [4], or using β-amylase treatment, etc. [5]. Other thermomechanical treatments, such as steam-jet cooking, extrusion, high hydrostatic pressure, and wet-heat processing, have also been employed [6]. These methods are, however, time-consuming and labor-intensive and require the use of potentially harmful solvents and chemicals/enzymes. The green synthesis proposed in this study has a significant advantage over the other methods in terms of environmental impact, accuracy, and convenience. Moreover, there are no high temperature/pressure treatments involved. Also, this method is fast, and the complexes can be prepared without the use of any external chemicals/enzymes [7].

Hydrogels are three-dimensional polymeric structures that are water-swollen and that consist of covalent bonds formed by the reaction of one or more comonomers, association bonds such as van der Waals interactions or hydrogen bonds between chains, and physical cross-links caused by chain entanglements. There have been reports of tremendous applications of hydrogels, including their use as biocompatible material in wound healing as well as in the formulation of electrets [8,9]. Starch hydrogels have drawn a lot of interest because of their affordability, reusability, stability when stored, biodegradability, and biocompatibility. A starch hydrogel is an amphiphilic cross-linked adsorbent that can be employed in a range of applications [10]. Different studies have reported different methods of producing starch hydrogels using physical or chemical processes to form strong and stable networks of the hydrogel [11]. Although the chemical approach may produce hydrogel with higher mechanical strength, the main drawbacks of this synthesis method are safety concerns, extensive processing times, and high energy consumption [12].

Significant emphasis has been placed on increasing the content of resistant starch (RS) in starch-based products due to RS’s health benefits against diabetes and obesity [13]. Additionally, active research is being conducted on starch derivatives with low glycemic indices, which show promising potential for use in the food and beverage industries [14].

The influence of starch hydrogel production on the physical properties that affect starch digestion is a significant factor. Therefore, the utilization of green synthesis, specifically the simple gelatinization of the amylose–lipid nanomaterial (ALN), may have a profound impact on the structure and digestibility of the hydrogels, leading to the production of the amylose–lipid nanomaterial. This material has the potential to be used as a low-glycemic gelling and functional food ingredient with health benefits for people with diabetes or other metabolic disorders. To test this hypothesis, the current study aims to produce amylose–lipid complexes through high-speed homogenization (green method), isolation of the ALN, and their structural/functional analysis. The study further explores the formulation of low-glycemic index hydrogel obtained from ALN along with rheology and anthocyanins retention capacity measurements.

## 2. Results

### 2.1. Structural Characteristics of Isolated ALN

The functional groups present in native starch and ALN were characterized using FT-IR spectroscopy to determine the effect of starch–fatty acid reactions (Figure 1). The control sample displayed an absorption band at approximately 3350 cm^−1^, which corresponds to the presence of the OH functional group. An absorption band around 2950 cm^−1^ indicated the presence of methylene/methine groups, while an intense band around 1000 cm^−1^ was observed, which arose from C-O strong stretching vibrations. However, there was no absorption detected for the presence of carbonyl groups. Conversely, all samples of the ALN showed a significant reduction in OH absorption and an increase in methylene group absorption probably because of the integration of long-chain fatty acids. The IR spectra of nanomaterial obtained from mixtures of lauric acid nanomaterial LA(NM) and stearic acid nanomaterial SA(NM) showed two distinct absorptions, one at 2900 cm^−1^ for CH_3_ and the other at 2800 cm^−1^ for CH_2_. The presence of the ester carbonyl group in ALN was indicated by three characteristic bands at 1700, 1200, and 1016 cm^−1^. The absorption band at 1016 cm^−1^ in ALN suggested the presence of amylose–fatty acid C-O functionality. Compared with previous results [15], no changes in the structures of the nanocomposite materials (SA(NM) and LA(NM)) were observed in the IR spectrum. However, the palmitic acid nanomaterial (PA (NM)) had a very low absorption of esterified moieties, which could be due to weak complexation or its susceptibility to enzymatic hydrolysis during isolation.

To evaluate the morphological changes, scanning electron microscopy (SEM) provides a comprehensive view of concerned changes in the shapes and structures of materials, especially nanoscale materials. Figure 2A–D shows SEM images of the control sample as well as the resulting complexes (ALNs). SEM scans indicate sizes ranging from 50 to 200 nm and from 100 to 500 nm for the control sample and all ALNs, respectively (with 0.8% fatty acids and 6% starch). Yan et al. [16] reported a 500 nm mean size for *Cyperus esculentus* starch–palmitic acid complex nanoparticles (with 0.4% palmitic acid to 4% starch) and reported an increase in particle sizes from 500 to 567.2 nm when hydrolysis time was extended, suggesting an increase in amylose content. The larger size of the amylose–lipid nanomaterial may have a slower degradation rate, which indicates sustained release of encapsulated molecules [17]. The SEM showed a granular spherical round shape of our control sample having sphere crystallites on its surface. The control sample exhibited granular structures with spherical, round-shaped spheres at the surfaces, with sphere crystals [18]. Guan et al. [19] found that the size and shape of starch granules have a significant impact on determining the degree of substitutions, particularly in the crystalline regions of the starch helical structures that facilitate the esterification of OH groups. The SEM images of SA-NM revealed that the granular structures of starch were no longer present owing to the binding of the fatty acid, resulting in the appearance of a solid, unbroken matrix structure of SA(NM). It is proposed that the mechanism involved in this process entails the breaking of hydrogen bonds in the helical structures of starch by stearic acid. This leads to the appearance of an intact and strong matrix structure [20]. This complex constitutes a smaller number of crakes and lamellae while no spherulites have been seen, which indicates a strong incorporation of the acid. The SEM image of PA(NM) (B), however, shows a granular structure without any matrix composition. It seems that the complex has somehow been affected by the enzymatic hydrolysis process and the fatty acid has not been fully incorporated, which can also be observed in the IR spectrum of PA(NM). Figure 2D corresponds to the SEM of LA(NM), which shows an intact solid matrix structure with a larger surface area having spherulites on its surface. The smoothness of the matrix surface is an indication of the formation of nanoscale material as well as the full incorporation of acid in the starch molecules [21]. SEM images of hydrogels formulated from each ALN show the morphology of amylose–lipid nanocomposite materials. PA hydrogel appears spherical with polydispersed particle size varying approximately from 200 to 350 nm in diameter with relatively smaller pores, approximately 50–100 μm sizes. LA hydrogels displayed larger spherical particle sizes varying approximately from 300 to 600 nm and relatively larger pores, approximately 200–250 μm sizes. SA hydrogel has a distinct morphology from other samples, with hard clumps of amylose–lipid networks firmly packed with a smaller number of pores having approximately 100–250 μm sizes. Its shape could be attributed to the hydrogels being stored at low temperatures prior to image analysis [22].

The XRD of all the samples was carried out to determine the relative crystallinity in the ALN (Figure 3a). The XRD of the control sample showed various peaks at 2Ɵ = 15.44°, 17.92°, 18.49°, and 23.09° as an indication of the C-type pattern, dissimilar to the B-type pattern due to the absence of a peak around 5°. A very small peak observed around 20.15 ° is the amylose–lipid characteristic peak, which is an indication of a residual lipid already incorporated in the control [23]. All the other treated samples (LA(NM), PA(NM), and SA(NM)) presented a V-type XRD pattern at 2Ɵ = 12.69°, 18.16°, 20.17°, and 22.18°, which corresponded to partial crystallinity type 1 [24]. The observed patterns at 20.17° and 22.18° are consistent with starch–acid complexes, which may be a consequence of fatty acid crystals being bound and trapped within amylose helices [25]. The relative crystallinity of ALNs has been shown to increase with the subsequent addition of longer-chain-length fatty acids (Figure 3a). The crystallinity increased from LA to PA (chain length increased from C-12 to C-16) due to the appearance of strong peak in XRD of PA(NM) at 22.18° [26] while it kept increasing from PA to SA (chain length increase from C-16 to C-18). It is possible that some of the peaks seen in PA and SA are a result of retrograded starch chains and starch–PA/SA complexes. In conclusion, the incorporation of fatty acids resulted in a significant increase in the diffraction peaks at 22.18°, which is indicative of the typical V-type pattern observed for amylose–lipid complexes, compared with the control sample. Furthermore, the diffraction peak around 5° observed in the amylose disappeared in the starch–fatty acid complexes [23].

The thermal and degradation properties of control and nanomaterial complexes were investigated via TGA (Figure 3b). The initial weight loss at temperature ranges from 100 to 125 °C may have been due to the evaporation of absorbed water molecules [26]. All kinds of samples, such as SA(NM), PA(NM), and LA(NM), including a control sample of starch, exhibited the same kind of behavior and showed a thermal event below 125 °C. The control and nanomaterial complexes both showed three-stage weight losses below 700 °C. The first major loss occurred due to the evaporation of water in the range of 35–125 °C, which is usually observed near 100 °C. The second major weight loss was found to be a major one with maximum percent loss for both the control and nanomaterial complex in the temperature range of 270–460 °C. The remaining weight loss was observed in the range of 460–640 °C [15]. These other two weight losses occurred due to the decomposition of starch amylose and amylopectin in the control and nanomaterial complexes [27]. Moreover, it is possible to see in the TGA curves that nanocomposite material complexes exhibited more stability compared with the control. The observed sample variance could be attributed to the possible inclusion of fatty acids within the starch structures. LA(NM) showed maximum stability because of its larger incorporation inside starch helices due to the short chain, which ultimately formed stronger ester linkages [28]. Also, the crystallinity of the resulting ALN could enhance the stability as has been observed in SA(NM) where the large-chain stearic acid provides extra crystallinity to the material. PA(NM) was not stable compared with both SA and LA(NM).

### 2.2. Complexing Index, Solubility, and Swelling Power of ALN

The complexing index (CI) was determined for isolated nanomaterials as shown in Table 1. In our previous study, the CI values for the mixture of starch–stearic acid, starch–palmitic acid, and starch–lauric acid were reported as 53.37, 24.16, and 38.26, respectively [26]. However, in this study, the isolated nanomaterial from the same mixtures shows CI values of 45.38, 29.94, and 40.9, respectively. The same scattered effect has been observed as previously, with a slight increase or decrease in values. These results indicate that mixtures were fully saturated with a starch–fatty acid mixture with a fixed ratio under high-speed homogenization. The CI values clearly demonstrate that the isolation process of nanomaterials through enzymatic hydrolysis has no promising effect on the complexion abilities of starch, while a slight increase in CI values may be attributed to an increase in the concentration of complexes after the removal of unbounded starch from the mixtures. The solubility (S) and swelling powers (SP) of the nanomaterial were observed to decrease compared with the control sample. The results of S and SP for nanomaterial in this study are similar to the already reported starch–fatty acid mixture [15]. It is obvious that such properties of starch remarkably depend upon the addition of fatty acid. The values of S and SP show that there is no adverse effect of the isolation process on the nanomaterial from the starch–fatty acid mixture and all the bonds are intact between lipids and starch as earlier noted in CI values.

### 2.3. In Vitro Digestibility and GI Values

In vitro digestion of the control sample and isolated nanomaterial is shown in Table 2. Various parameters for the digestion curves show significant differences. All the curves show nonlinear behavior and have different reaction constant (Kt), intercept, and slope values. The calculated glycemic index value (GI) was obtained by dividing the total hydrolysis curve (AUC) of the sample by the hydrolysis curve of the white bread curve. Although no significant difference was observed in GI values of all nanomaterials tested, GI values were significantly reduced in comparison to the control sample in all the nanomaterials.

The results indicate that the homogenization of any fatty acid in this study with native starch can decrease the GI values. It has been reported that food having higher GI value can be digested in the intestine rapidly and can increase the glucose level sharply in the blood [29]. The lower values of GI in the presence of fatty acids reveal that homogenization of the starch–fatty mixture at high speed reduced the glycemic response of the nanomaterials in digestion. Generally, diets that are low in saturated fatty acids and high in carbohydrates are recommended for insulin resistance and diabetes. Many studies have been reported on the relationship between GI and type II diabetes along with the protective effect of low GI food in cardiovascular disease as well as in cancer [30]. The findings for GI in this study indicate that using such nanomaterials in food processes and micronutrient delivery can help reduce obesity, substitute blood fats, and provide a healthy meal for type II diabetic patients [31].

### 2.4. Total Starch (TS), Resistant Starch (RS) and Digestible Starch (DS)

Dry weight basis calculations were used to determine the supposed DS by taking the TS (100%) and the RS (in %) obtained according to the formula:DS = TS − RS(1)

After calculating the in vitro digestibility, it was found that the TS remained higher in the control sample followed by the ALN. The in vitro digestibility measurements showed a significant decrease in DS for ALN compared with dry weight measurements (Figure 4).

RS usually refers to the starch that remains unchanged through the digestion track and somehow shows properties of soluble fiber. Some of the important functions of RS include the reduction in blood glucose levels and appetite as well as improvement in insulin sensitivity [32]. The calculated RS values for nanomaterials were observed to be significantly higher compared with the control sample (Figure 4b). The higher RS values were observed in LA(NM) (88.15%), followed by SA(NM) (62.99%) and PA(NM) (62.56%), respectively. RS has been divided into five classes (RS1, RS2, RS3, RS4, and RS5) on the basis of their nature and granule structure [33]. Naturally, all types of starch contain RS as shown in Figure 4b; however, the increase in RS values for isolated ALN indicates that such type of starch falls in the RS4 category, where the improvement was achieved via a modification technique [34].

Our results reveal that the obtained ALN, i.e., SA(NM), PA(NM), and LA(NM), have higher RS values and could be used as food ingredients in various food development processes. It has been reported that the digestibility of starch both in in vivo and in vitro depends on the starch source and starchy food processing [34]. Furthermore, cereal starch digestion is quite easier in comparison to starch of root/tubers. A significant decrease was observed in amounts of DS (Figure 4c) for SA(NM), PA(NM), and LA(NM), respectively. These results indicate that the addition of fatty acids to starch under high-speed homogenization may shift a rapidly digestible starch into slowly digestible starch, which is usually linked to stable glucose metabolism, diabetes management, and improved satiety [35].

### 2.5. Rheology

The rheological properties of each hydrogel were characterized by frequency sweep analysis as measured by a rheometer at a fixed strain of 10% and 25 °C temperature. The strain and angular frequency (ω) ranges of 1% and 0.1–100 rads^−1^, respectively, were used, with a standard normal force of 0.5 N. The viscosity of each hydrogel and the G′ and G″ are illustrated in the plot in Figure 5. From the viscosity curve, all formulated hydrogels exhibit viscoelastic properties as indicated by the decreasing trend in viscosity as the frequency increases. Although hydrogels exhibit high viscosity at low frequencies, SA hydrogel demonstrates the highest resistance to flow, followed by LA and PA hydrogels (*p* > 0.05), as evidenced by the distinctive shapes of the viscosity curves depicted in Figure 5a. This can be attributed to the relaxation time of the hydrogel, which is closely associated with the molecular mobility of the starch–stearic acid network. The unique unbroken matrix structure of the SA nanomaterial, as evidenced by the SEM scans of the nanomaterial, can be traced as the underlying reason for this phenomenon. However, as the shear rate increases or the frequency increases, the viscosity of the hydrogel diminishes, owing to the relaxation of internal forces within the material. This viscosity behavior is indicative of non-Newtonian behavior and is characteristic of all starch-based hydrogels, including nanocomposite hydrogels, as previously reported by other researchers [36]. Viscosity curves could provide insights into how hydrogels will behave during processing and this characteristic could prove valuable in applications such as formulation of gelling food products where hydrogels are best needed to maintain structural integrity.

The storage modulus (G′) curve shows an ascending trend with frequency, indicating that all hydrogels display solid-like behavior with a constant resistance to deformation. This trend further suggests that all hydrogels become more elastic with frequency. Typically, an ideal hydrogel demonstrates a distinct plateau on the G′ curve, which extends to at least seconds in this region. In Fig 5B, hydrogels show a plateau from 4 rads/s to 10 rads/s at high frequency, indicating increased stiffness and reduced ability to flow, consistent with the behavior of an ideal hydrogel. This phenomenon could be attributed to the physical cross-link networks of short amylose and fatty acid polymers, which reconfigure to store and release applied energy [37]. Notably, the SA hydrogel shows higher storage modulus values at lower frequencies, signifying its more solid-like/stiff morphology relative to other hydrogels. Additionally, the storage modulus (G′) of hydrogels surpasses the loss modulus (G″) over the entire frequency range, implying that these nanomaterials are predominantly elastic and can store energy when deformed. The peak of the loss modulus of all hydrogel samples is reached at 5 Hz, which is the gel point above which the hydrogels behave more like a liquid and below which they behave more like a solid, consistent with prior research on hydrogels formed by mixing starch and fatty acids [38]. This characteristic suggests that the gelation of these observed hydrogels can be controlled to yield a stable gel structure within precise timeframes and under specific processing conditions. This precision ensures the attainment of the desired texture and stability in the final product. Understanding the rheological behavior of these hydrogels is crucial when utilizing these hydrogels in food formulations as it could impact the texture, mouthfeel, and sensory attributes of the final food product. Also, the elasticity and ability of these hydrogels to store energy (i.e., G′ and G″) are crucial in food texture modifications. Quality hydrogels must demonstrate a dual nature of both viscosity and elasticity. Based on our findings, the SA hydrogels, characterized by high viscosity, have the potential to impart stability, texture, and an appealing mouthfeel when incorporated into food products. Furthermore, all hydrogels examined in this study exhibit a valuable trait—thixotropy. This means that under increased shear stress, they become less viscous, facilitating easy pouring, while returning to their original viscosity at rest. Such a feature is highly sought after in the food industry for ensuring products maintain a stable texture when not in use.

### 2.6. Retention of Anthocyanin

Anthocyanins are of interest in this study due to their susceptibility to stability-altering factors, including pH, light, temperature, and processing conditions [39]. Therefore, it is imperative to investigate their behavior, encompassing absorption and retention, within food matrices, specifically hydrogels, under varying pH conditions. This study centers its focus on anthocyanins due to their recognized potential health benefits and antioxidant characteristics. The absorption of anthocyanins by hydrogels can be influenced by pH, which prompted our study to evaluate the effect of pH on the anthocyanin absorption capacity of formulated amylose–lipid nanocomposite hydrogels. It is widely acknowledged that anthocyanins exhibit greater stability and absorption efficiency under low pH conditions, while high pH values can lead to a reduction in absorption due to the decreased stability of anthocyanins, as reported by Enaru et al. [40]. We measured the absorption capacity of formulated hydrogels at pH 2, 4, 5, 7, and 8 (Figure 5d). According to our findings, the absorption patterns of all hydrogels, including the PA, SA, and LA hydrogels, were similar. The largest absorption capacity was seen at the lowest pH of 2, which may have been due in part to anthocyanins’ improved stability in acidic environments, as reported in past studies by Ghareaghajlou et al. [41]. Notably, the SA hydrogel showed the highest adsorption capacity, followed by the LA hydrogel and PA hydrogel. This observation can be attributed to the high viscosity of the hydrogel, which is typically linked to its retention capacity within the polymeric structure. The ability of hydrogels to absorb water arises from the functional groups attached to the polymeric backbone and cross-links between network chains, as previously reported [41]. The high viscosity recorded for the SA hydrogel during rheology measurement and the solid amylose–lipid matrix structure that makes up the hydrogel is, therefore, likely to be responsible for its high absorption capacity. Furthermore, as shown in Figure 5d, the uptake capacity of anthocyanins by hydrogels decreases with increasing pH. These results are in agreement with those recorded in other studies [38]. Analysis of anthocyanin encapsulation and retention by amylose–lipid-based hydrogels demonstrates the effect of pH and degree of physical cross-link networks of short amylose and fatty acid polymers on the uptake capacity of anthocyanins.

## 3. Materials and Methods

The three fatty acids (lauric acid; palmitic acid; and stearic acid) were purchased from Aladdin Chemistry Co., Ltd. (Merck, Rahway, NJ, USA). The native cornstarch was obtained from the local market. The starch was defatted with *n*-hexane to remove any residual lipids and was stored at 4 °C in sealed tubes for further experiments. Analytical-grade chemicals were used throughout all the experiments.

### 3.1. Preparation of Starch–Fatty Acid Complexes

To create starch–fatty acid complexes, maize starch (6 g) was mixed with 75 mL of distilled water, and stirred for one hour, and 0.75 g of each stearic acid, palmitic acid, or lauric acid (dissolved in ethanol (15 mL)) was added without heating or enzymatic treatment. The mixture was then homogenized using a homogenizer at 30,000 rpm for 10 min. A control sample was prepared similarly, except that no fatty acids were added. The resulting mixtures were kept at room temperature, followed by the addition of 10 mL of ethanol to remove unbound fatty acids. The mixtures were centrifuged at 10,000 rpm, and the resulting precipitates (LA complex, PA complex, and SA complex) were each collected and weighed (4.7 g). The precipitates were subsequently subjected to freeze-drying for further analysis [26].

### 3.2. Isolation of Amylose–Lipid Nanocomposite Material (ALN) from Starch Fatty Acid Mixtures

The freeze-dried samples of each complex were stirred in distilled water at 50 °C and 90 °C for 10 s each, followed by 130 min of stirring at 75 °C. Thermostable α-amylase (0.5 mL) was added for hydrolysis. Ethanol (5 mL) was added to terminate the enzymatic reactions after 20 min, and the mixture was cooled at 25 °C. The hydrolyzed material was dispersed in distilled water and washed three times. The collected precipitate was oven-dried at 45 °C for 48 h. The yield obtained for each complex was 58–70.5% [42].

### 3.3. Fourier Transform Infrared Spectroscopy (FTIR)

Fourier transform infrared (FTIR) spectra of the samples were recorded using a Cary 660 FTIR spectrometer with ATR assembly from Agilent Technologies, Santa Clara, CA, USA. Spectra were collected at 4 cm^−1^ resolution, with 256 scans per sample, in the range of 600–4000 cm^−1^. The purpose of the analysis was to identify the functional groups present in the samples.

### 3.4. Scanning Electron Microscopy (SEM)

Observation of any physical changes in the surface morphology after cross-linking of both ALN and ALN-hydrogel samples was made from SEM images using a “TESCAN VEGA (LMU) instrument” at various magnifications, i.e., 20, 50, 100, 200, and 500 µm and a Quorum Sputter coater “SC7620 device”, which was used for gold-coating of our samples [22].

### 3.5. X-ray Diffraction (XRD)

The XRD experiments were performed to determine the relative crystallinity in complexes using Shimadzu (Tokyo, Japan), LabX XRD-6100. The instrument was operated at an accelerating voltage of 30 KV and a current of 30 mA using Cu-Ka radiation with a wavelength of 1.5406 Å. The measurements were scanned at 2θ range of 10° to 70°.

### 3.6. Thermal Gravimetric Analysis (TGA)

The thermal stability of the samples was evaluated using PerkinElmer thermal gravimetric analysis (TGA), Pyris-1, V-3.81, under N_2_ gas flow in the temperature range of 50–800 °C at a heating rate of 10 °C/min.

### 3.7. Complexing Index (CI)

The complexing index (CI) for all samples was determined following the method described by Tang and Copeland [43]. ALN samples in the amount of 0.4 g were dispersed in 5 g of distilled water and heated in a water bath at 60 °C until complete starch gelatinization was achieved. After cooling to room temperature and dilution, 0.5 mL of the resulting mixture was added to a solution of distilled water and iodine solution. The absorbance was measured at 690 nm, and CI values were calculated using Equation (2).
(2)$$\;\mathrm{Complexing}\;\mathrm{Index}\;\left(\mathrm{CI}\right)=\frac{\mathrm{Abs}\;\left(\mathrm{Ref})-\mathrm{Abs}\;(\mathrm{ALN}\right)}{\mathrm{Abs}\;\left(\mathrm{Ref}\right)}$$

Abs (Ref) = absorbance of a control sample; Abs (ALN) = absorbance of isolated nanomaterials.

### 3.8. Swelling Power (SP) and Solubility (S)

The swelling power and solubility of ALN were determined as per the modified method of Ijaz et al. [15]. A 4% (*w*/*w*) sample was heated in a water bath at 60 °C for 30 min, cooled to 20 °C, and centrifuged at 15,000 rpm for 10 min. The weights of both supernatant and precipitate were determined and used in the calculation of “solubility and swelling power” (H_2_O absorbed (g)/dry sample (g)) with the following Equations (3) and (4).
(3)$$\mathrm{Solubility}\;\left(S\right)=\frac{\mathrm{Weight}\;\mathrm{of}\;\mathrm{dried}\;\mathrm{supernatant}}{\mathrm{Dried}\;\mathrm{weight}\;\mathrm{of}\;\mathrm{amylose}-\mathrm{lipids}\;\mathrm{nanomaterial}}\times 100$$
(4)$$\;\;\;\mathrm{Swelling}\;\mathrm{power}\;\left(\mathrm{SP}\right)=\frac{\mathrm{Weight}\;\mathrm{of}\;\mathrm{precipitate}}{\mathrm{Dried}\;\mathrm{weight}\;\mathrm{of}\;\mathrm{amylose}-\mathrm{lipids}\;\mathrm{nanomaterial}}\times 100$$

### 3.9. In Vitro Digestibility and Glycemic Index (GI) of Isolated ALN

The in vitro digestibility of ALN was determined using 250 mg of each sample, with a modified version of the method described by McCleary et al. [44]. Amyloglucosidase and glycerol aqueous solution were mixed and stored in the refrigerator. Pancreatin was stirred in a sodium maleate buffer before adding amyloglucosidase solution. After centrifugation, the supernatant was collected and mixed with gelatinized ALN. The mixture was shaken and centrifuged, and the supernatant was collected. The remaining starch was hydrolyzed by adding KOH and stirring. Sodium acetate buffer and amyloglucosidase in glycerol were added, and the mixture was stirred. Glucose in the supernatant was analyzed and the glycemic index value (GI) was calculated by dividing the AUC by the hydrolysis curve of white bread. The GOPOD method was used to quantitatively analyze glucose in the supernatant.

### 3.10. Total Starch

To determine the total starch in the control and ALN samples, a modified method based on Goñi et al. [45] was used. The samples were ground and dispersed in 2 M KOH, then hydrolyzed with amyloglucosidase, and incubated at 60 °C for 45 min. After centrifugation, the glucose concentration was determined and converted into starch content using a factor of 0.9. Color absorption was measured at 450 nm. Approximately 25–35 mg of each sample was used and the solubilized starch was vigorously shaken at 25 °C for 30 min. The glucose oxidase-peroxidase kit was used for glucose concentration determination in the supernatant.

### 3.11. Resistant Starch and Digestible Starch

To determine resistant starch (RS), 100 mg of the sample was incubated with pepsin-containing solution for 60 min at 40 °C to eliminate proteins. The starch was then hydrolyzed with α-amylase for 16 h at 37 °C, and the samples were centrifuged. The remaining residue was analyzed for starch content using amyloglucosidase. The amount of digestible starch (DS) was calculated by subtracting RS from the total starch (TS) content [45].

### 3.12. Preparation of Hydrogel

A liquid mixture of ALN (8g/L) prepared in a 15 mL tube was heated for 30 min at 90 °C with constant mixing in a water bath where the sample gelatinized and produced a gel. The texture of the gel was appropriate and reproducible; hence, it was poured into the specific particularized holder of 1 × 2 cm, capped with a glass plate. The gel was stored at 4 °C for characterization and further testing.

### 3.13. Rheological Measurements

Rheological investigations were conducted on a PHYSICA M CR 300 apparatus (Stuttgart, Germany). To ascertain the storage modulus (G′) and the loss modulus (G″) and their dependence on angular frequency (ω), a frequency sweep analysis was carried out at a fixed strain (10%) and a frequency range of 0.1–10 Hz at a temperature of 25 °C. During experimentation, the strain and angular frequency ranges of 1% and 0.1–100 rad/s, respectively, were used with a standard normal force of 0.5 N. The data were acquired using the MCR300 software (1T-1S-V2).

### 3.14. Encapsulation and Uptake of Anthocyanin

Anthocyanin–hydrogel complexes were prepared according to the method described by Donghwan and Jaeyuns (2021) [46]. A total of 1 mL of 10 mg/mL of anthocyanin solution was added to a solution of 10 mg of particles of dried hydrogel already mixed in 9 mL of buffer at pH 2, 4, 5, 7, and 8. The mixture was gently stirred for 4 h and subsequently centrifuged at 8000 rpm for 10 min. Anthocyanin content was calculated as follows (Equation (5)).
(5)$$\Gamma =\frac{m_{\mathrm{add}}-{\;C}_{\mathrm{prot}}\times {\;V}_{\mathrm{supernatant}}}{m_{\mathrm{dry}\;\mathrm{gel}}}$$
where V_supernatant_ represents the volume of the supernatant (mL); C_prot_ is the anthocyanins concentration in the supernatant (mg/mL); m_add_ is the total anthocyanins added (mg); and m_dry gel_ is the mass of added dry hydrogel (g).

### 3.15. Statistical Analysis

The experiments were carried out in triplicate and the resulting data were subjected to statistical analysis using one-way ANOVA. The means were compared using the Duncan multiple range test with a significance level of *p* < 0.05.

## 4. Conclusions

In conclusion, the study produced amylose–lipid nanocomposite (ALN) materials via simple homogenization and hydrolysis procedures. The materials were characterized using various techniques and formulated into hydrogels for future applications. Overall, the results show that ALN materials have distinct structural and chemical characteristics, and, hence, the hydrogel formulation makes them promising for a variety of applications, such as medicine administration, food packaging, and other industrial purposes. This study’s findings contribute to the expanding knowledge of the characteristics and potential applications of amylose–lipid nanocomposite materials. Further research is needed to thoroughly investigate the potential of ALN hydrogels in many fields and to upscale the green synthesis and processing procedures for specific applications. In conclusion, the anthocyanin uptake capacity results of ALN hydrogels presented SA hydrogels with the best absorption capacity.

## Figures and Tables

**Figure 1 molecules-28-07154-f001:**
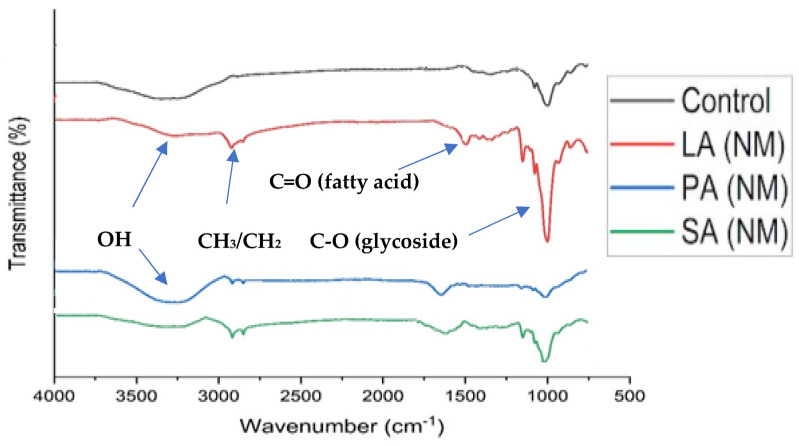
FTIR spectra of control and ALN. SA(NM) spectrum for ALN isolated from maize starch-stearic acid mixture, PA(NM) spectrum for ALN isolated from maize starch-palmitic acid mixture, and LA(NM) spectrum for ALN isolated from maize starch-lauric acid mixture.

**Figure 2 molecules-28-07154-f002:**
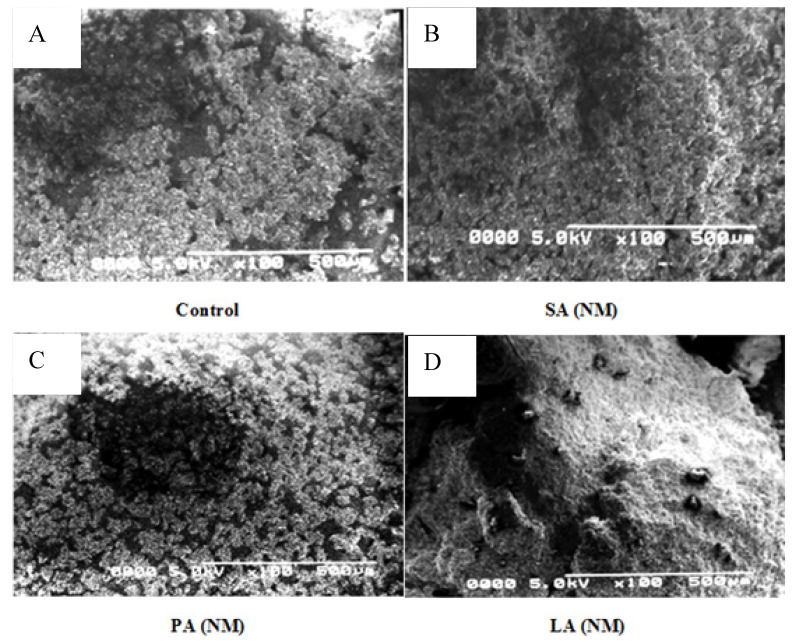
SEM images of ALN and ALN hydrogel. (**A**) = (control) maize starch, (**B**) = ALN isolated from the stearic acid mixture, (**C**) = ALN isolated from the palmitic acid mixture, and (**D**) = ALN isolated from the lauric acid mixture, (**E**) = PA hydrogel, (**F**) = LA hydrogel, and (**G**) = SA hydrogel (100 μm).

**Figure 3 molecules-28-07154-f003:**
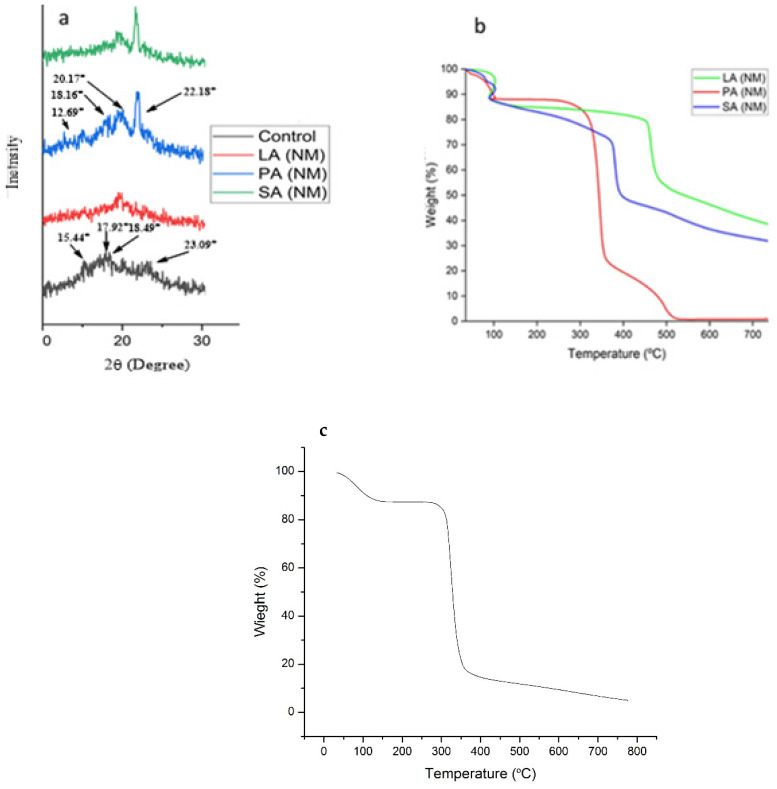
(**a**) XRD spectra of control and ALN. SA(NM) spectra for ALN isolated from the stearic acid mixture, PA(NM) spectra for ALN isolated from the palmitic acid mixture, and LA(NM) spectra for ALN isolated from the lauric acid mixture. (**b**) TGA curve of ALN. SA(NM) curve for ALN isolated from the stearic acid mixture, PA(NM) curve for ALN isolated from the palmitic acid mixture, and LA(NM) curve for ALN isolated from the lauric acid mixture. (**c**) TGA curve of the control.

**Figure 4 molecules-28-07154-f004:**
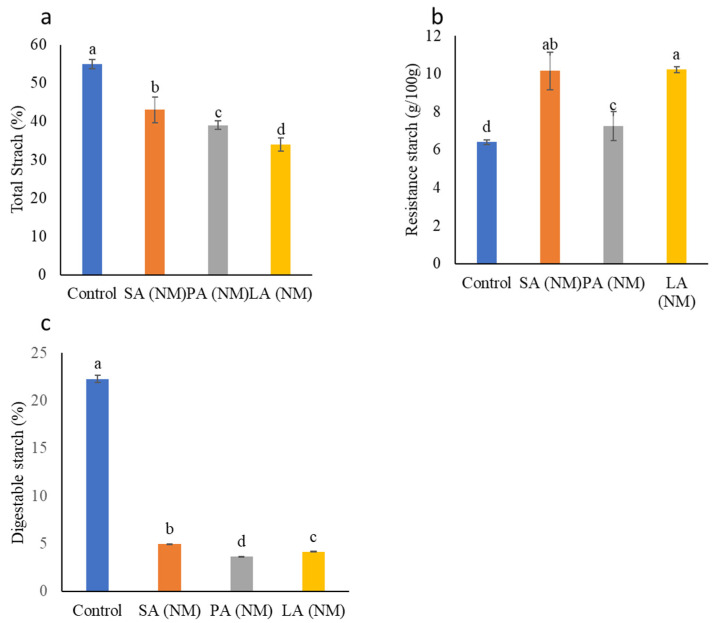
Amount of total starch (**a**), resistance starch (**b**), and digestible starch (**c**) of ALN estimated on the dry-weight basis. Different characters ^a,b,c^ and ^d^ represent significant differences (*p* ≤ 0.05). The letters ^a,b,c^ and ^d^ each signify statistically significant distinctions (*p* ≤ 0.05) These comparisons are articulated with regard to the observed differences in significance between the control and experimental outcomes.

**Figure 5 molecules-28-07154-f005:**
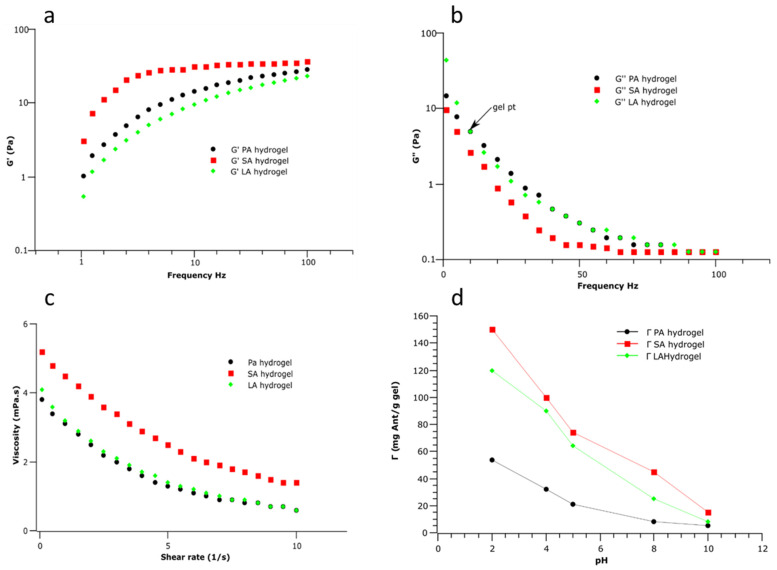
Top—Storage modulus, loss modulus, and flow curves of (**a**–**c**) of PA, SA, and LA hydrogels as a function of frequency performed at a fixed strain of 10% and 25 °C temperature maintained. Bottom—Flow behavior of PA, SA, and LA hydrogels. (**d**) Anthocyanins (Ant) uptake capacity Γ (mg and/g gel) of amylose–lipid nanocomposite hydrogels as a function of pH.

**Table 1 molecules-28-07154-t001:** Complexing index, solubility, and swelling power of ALN.

	Control	SA (NM)	PA (NM)	LA (NM)
Complexing index	-------	45.38 ± 0.004 ^b^	29.94 ± 4.960 ^c^	40.9 ± 0.000 ^a^
Solubility	84.5 ± 0.04 ^a^	47.5 ± 0.009 ^d^	79 ± 0.078 ^b^	59.51 ± 0.005 ^c^
Swelling power	13.75 ± 0.001 ^a^	8.25 ± 0.003 ^c^	5.75 ± 0.007 ^d^	10.75 ± 0.002 ^b^

Note: Values shown are mean ± SEM, No. of experiments = 3. The different characters ^a,b,c,d^ represents significant difference (*p* ≤ 0.05). The letters ^a,b,c^ and ^d^ each signify statistically significant distinctions (*p* ≤ 0.05) These comparisons are articulated with regard to the observed differences in significance between the control and experimental outcomes.

**Table 2 molecules-28-07154-t002:** Complexing index (CI), solubility (S), swelling power (SP), glycemic index (GI) values, and different parameters of GOPOD during in vitro digestibility of control and isolated ALN.

GOPOD Parameters	Control	SA (NM)	PA (NM)	LA (NM)
Slope	0.00089 ± 0.0007	0.00073 ± 0.0002	−0.00227 ± 0.0002	−0.00019 ± 0.0002
Intercept	−1.04558 ± 0.0001	−0.95603 ± 0.000	−0.57345 ± 0.000	−0.99518 ± 0.000
K_t_ (h^−1^)	−0.00205 ± 0.000	−0.00168 ± 0.000	0.00523 ± 0.0004	0.000426 ± 0.000
T_1/2_ (h)	−337.938 ± 0.000	−412.489 ± 0.000	132.502 ± 0.710	1625.087 ± 1.777
C_o_ (µg/mL)	0.090036 ± 0.005	0.110655 ± 0.000	0.267021 ± 0.0003	0.101117 ± 0.0008
Vd (L)	555.3325 ± 0.0001	451.8567 ± 0.001	187.251 ± 0.0001	494.4761 ± 0.007
Clearness (L/h)	−1.13881 ± 0.0000	−0.75914 ± 0.000	0.979344 ± 0.001	0.210864 ± 0.000
AUC 0 to t	0	0	0	0
AUC 1 to t	19.61 ± 0.0001	23.76 ± 0.0002	32.7 ± 0.0000	17.02 ± 0.0000
AUC t to Infinity	−1.06469 ± 0.000	−1.46821 ± 0.000	0.455695 ± 0.0002	4.103756 ± 0.0000
GI (%)	76.86 ± 1.271	40.29 ± 0.001	40.58 ± 0.001	40.26 ± 0.000

Note: Values shown are mean ± SEM, No. of experiments = 3. Reaction constant (K_t_), half of the hour (T_1/2_ (h)), the concentration of the sample (C_o_ (µg/mL), the volume of the solution (Vd (L)), the area under the curve (AUC), and glycemic index (GI).

## Data Availability

Data available with the first author.

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
