# Peer review of "Green Synthesis of Low-Glycemic Amylose–Lipid Nanocomposites by High-Speed Homogenization and Formulation into Hydrogel"

_molecules, 2023, doi:10.3390/molecules28207154_

Round 1
Reviewer 1 Report
Dear Authors,
I have received the manuscript “Green synthesis of low-glycemic Amylose-Lipid Nanocomposites by high-speed homogenization and formulation into nanocomposite hydrogel” to be reviewed for the Molecules (ISSN 1420-3049) Journal. This is an original study that investigates amylose-lipid nanomaterials and their respective hydrogels aiming to achieve stable low-glycemic food products. I consider the presented results promising and relevant to the field. The acquired Low Glycemic Index materials can be helpful in meeting demands in food and health technologies. However, in general, the results were not presented properly. The images, graphs, and tables can be improved for a better understanding by the reader. Furthermore, the discussion section lacks contextualization regarding how the achieved properties may be important for the proposal. For example, in section 2.4, the authors discussed the results with proper contextualization. The same idea could have been applied to other sections, mainly 2.5 and 2.6. Below are specific comments regarding my point of view of the manuscript. The authors should view the comments as positive criticism and adequately address them in the review. I recommend the paper be reconsidered after Major Revisions.
*(L=lines of the manuscript)
1. I understood the need for specific keywords in the title. However, the title of the manuscript is too long and cites "nanocomposite" twice. Please consider reviewing the title of the manuscript (L1-3).
2. Still about the title. “Green synthesis” is used in the title but mentioned only two times in the text. Why do the authors consider their synthesis “Green”? Which type of chemicals were avoided and replaced compared to similar reports in the literature? If this can indeed be considered a Green Synthesis, the authors should justify why.
3. The abstract is a short version of the paper. In this case, it should begin with a brief introduction of the idea (L26-36). Please review the abstract structure consulting previously accepted papers of the journal for inspiration.
4. In general, the introduction content is satisfactory. However, the authors can improve and restructure it. There is a long paragraph (L56-76) that can be divided into at least two, giving particular emphasis to the aim of the manuscript, as well as the innovative contribution.
5. The fatty acids abbreviations must be defined in the text before being cited in the figures. Also, choose an abbreviation pattern for the formulations (e.g., LA-NM in L88 or LA (NM) in L152)
6. The quality of Figures 1 and 2 (A, B, C, and D) should be improved.
7. Section 2.1. L92, which previous results? This section was discussed without any references.
8. Could the authors explain how they concluded the idea mentioned in L128-129 from SEM results?
9. Please review the SEM discussion about hydrogels (L130-138). More specifically, when the authors mention, “PA-hydrogel appears spherical with polydispersed particle size varying approximately from 200-350 nm…” (L131-132). The SEM images presented in this work lack resolution and clarity for such a conclusion.
10. Please add the Intensity (a.u.) as the y-axis in Figure 3A. Also, I would suggest restricting the diffractogram patterns to the region of interest discussed in lines 147-165 to help the reader visualize the peaks.
11. Please verify and standardize the significant figures in Table 1. Also, consider adopting scientific notation to facilitate understanding.
12. I suggest the authors separate Table 1 into two tables regarding each item of the discussion (2.2 L190 and 2.3 L208).
13. What do the letters a,b,c,d, and ab mean at the top of each bar in the graphs of Figure 4? The caption of this figure should be improved (L243).
14. L289 review the unit rad/s.
15. Review Figure 5C caption in the graph.
16. I consider the technical discussion of Rheology satisfactory. However, there is no contextualization. Why these rheological parameters are important for the proposed application? Why the food industry is interested in hydrogels with these characteristics? Where can the food industry apply these hydrogels?
17. Please justify and contextualize why anthocyanins were used in the study (L301).
18. The method 3.1 should be reviewed. There is no information about the amount of the components to prepare the starch-fatty acid complexes. The authors mentioned in L105 the final percentage of the components. Was not this supposed to be mentioned in the methods?
19. In L406, "ALN hydrogels presented the best absorption capacity," what does this sentence mean? Best absorption capacity compared to what? Please review the conclusion section to improve the insights and findings from the results.
Author Response
Reviewer #1
Dear Authors,
I have received the manuscript “Green synthesis of low-glycemic Amylose-Lipid Nanocomposites by high-speed homogenization and formulation into nanocomposite hydrogel” to be reviewed for the Molecules (ISSN 1420-3049) Journal. This is an original study that investigates amylose-lipid nanomaterials and their respective hydrogels aiming to achieve stable low-glycemic food products. I consider the presented results promising and relevant to the field. The acquired Low Glycemic Index materials can be helpful in meeting demands in food and health technologies. However, in general, the results were not presented properly. The images, graphs, and tables can be improved for a better understanding by the reader. Furthermore, the discussion section lacks contextualization regarding how the achieved properties may be important for the proposal. For example, in section 2.4, the authors discussed the results with proper contextualization. The same idea could have been applied to other sections, mainly 2.5 and 2.6. Below are specific comments regarding my point of view of the manuscript. The authors should view the comments as positive criticism and adequately address them in the review. I recommend the paper be reconsidered after Major Revisions.
*(L=lines of the manuscript)
- I understood the need for specific keywords in the title. However, the title of the manuscript is too long and cites "nanocomposite" twice. Please consider reviewing the title of the manuscript (L1-3).
Response: We appreciate your astute insight and we acknowledge your take. The title has been corrected based on your instructions. The new title is
“Green synthesis of low-glycemic Amylose-Lipid Nanocomposites by high-speed homogenization and formulation into hydrogel”
- Still about the title. “Green synthesis” is used in the title but mentioned only two times in the text. Why do the authors consider their synthesis “Green”? Which type of chemicals were avoided and replaced compared to similar reports in the literature? If this can indeed be considered a Green Synthesis, the authors should justify why.
Response: Most literature reports the chemical process of producing starch-fatty acid complexes, involving the solubilization of amylose in chemical solvents such as DMSO or KOH, followed by pH adjustment using HCL or enzymatic polymerization of glucose-1-phosphate. However, our research emphasizes homogenizing starch with fatty acids. This approach is not only promising for inducing complex formation and releasing amylose from swollen structures but has also been extensively elaborated upon in our previous work. Therefore, we used 'green synthesis' in our research title. To enhance flexibility and avoid the repetitive use of the phrase 'green synthesis,' we have removed it and rephrased the topic
- The abstract is a short version of the paper. In this case, it should begin with a brief introduction of the idea (L26-36). Please review the abstract structure consulting previously accepted papers of the journal for inspiration.
Response: The Abstract has been rewritten accordingly. We appreciate your suggestion. See page 1, line 26-41
- In general, the introduction content is satisfactory. However, the authors can improve and restructure it. There is a long paragraph (L56-76) that can be divided into at least two, giving particular emphasis to the aim of the manuscript, as well as the innovative contribution.
Response: The abstract has been adjusted and restructured according to your guidelines. Line 62-87, Page 2.
- The fatty acids abbreviations must be defined in the text before being cited in the figures. Also, choose an abbreviation pattern for the formulations (e.g., LA-NM in L88 or LA (NM) in L152)
Response: The LA-NM has been corrected throughout the MS. Line 99-108, Figure 1, page 3. 135, 137, 139 and so on highlighted as yellow.
- The quality of Figures 1 and 2 (A, B, C, and D) should be improved.
Response: Correction effected. Thank you for your suggestion
- Section 2.1. L92, which previous results? This section was discussed without any references.
Response: Reference cited. We appreciate your correction
- Could the authors explain how they concluded the idea mentioned in L128-129 from SEM results?
Response: The size measurements in the identified sentence were obtained from a software tool (ZEN -Carl Zeiss Efficient Navigation) for analysis of images obtained from SEM. The software could quantify and identify various features within images inputted including size, shape and other structures of particles.
- Please review the SEM discussion about hydrogels (L130-138). More specifically, when the authors mention, “PA-hydrogel appears spherical with polydispersed particle size varying approximately from 200-350 nm…” (L131-132). The SEM images presented in this work lack resolution and clarity for such a conclusion.
Response: Paragraph reviewed and corrections effected. The SEM images have been attached separately.
- Please add the Intensity (a.u.) as the y-axis in Figure 3A. Also, I would suggest restricting the diffractogram patterns to the region of interest discussed in lines 147-165 to help the reader visualize the peaks.
The Y axis has been labeled (intensity). We are sorry as we are unable to format the XRD diffractogram for the mentioned purpose.
- Please verify and standardize the significant figures in Table 1. Also, consider adopting scientific notation to facilitate understanding.
Response: Corrected
- I suggest the authors separate Table 1 into two tables regarding each item of the discussion (2.2 L190 and 2.3 L208).
Response: Table 1 has been separated into two tables.
- What do the letters a,b,c,d, and ab mean at the top of each bar in the graphs of Figure 4? The caption of this figure should be improved (L243).
Response: The different letters on same chart indicates significant differences at p<0.05. the caption is also modified.
- L289 review the unit rad/s.
Response: Reviewed and corrected. line 297, page 9.
- Review Figure 5C caption in the graph.
Response: Corrected
- I consider the technical discussion of Rheology satisfactory. However, there is no contextualization. Why these rheological parameters are important for the proposed application? Why the food industry is interested in hydrogels with these characteristics? Where can the food industry apply these hydrogels?
Response: Correction effected. Line 332-336. Page 10.
- Please justify and contextualize why anthocyanins were used in the study (L301).
Response: Thank you for your insightful suggestion. Justifications have been provided as instructed. Line 336-343, page 10.
- The method 3.1 should be reviewed. There is no information about the amount of the components to prepare the starch-fatty acid complexes. The authors mentioned in L105 the final percentage of the components. Was not this supposed to be mentioned in the methods?
Response: The amounts of reactants was added. See section 3.1, page 11, line 279-381.
- In L406, "ALN hydrogels presented the best absorption capacity," what does this sentence mean? Best absorption capacity compared to what? Please review the conclusion section to improve the insights and findings from the results.
Response: conclusion reviewed accordingly and corrections effected. Line 497-505, page 14.
Reviewer 2 Report
In this study, authors isolated and characterized Amylose-Lipid Nanocomposite mate- 26 rial (ALN) from starch and lipid mixtures containing stearic, palmitic, and lauric acids. The ALN was formulated into nanocomposite hydrogels to evaluate its anthocyanin absorption capacity. Furthermore, this study evaluates the anthocyanin encapsulation and retention by ALN-based hydrogels, highlighting the impact of pH and physical crosslink networks on the uptake capacity. The results are interesting and authors are advised to address the following issues.
1. The introduction section is lack of scientific background. Authors are advised to add an additional paper on hydrogel: https://doi.org/10.3390/jfb14060320, https://doi.org/10.1016/j.ijbiomac.2023.125754.
2. Section 3.15: The experiments were carried out in triplicate and the resulting data was subjected to statistical analysis using one-way ANOVA. The means were compared using the Duncan multiple range test with a significance level of P < 0.05. However, no statistical analysis results can be found in the figures.
3. Authors please add a blank before and after the “±”. Please remove “title” from the title. XRD, SEM, TGA are common characterization methods, which are not advised to add in the keywords list.
4. The IR spectra of nanomaterial obtained from mixtures LA-NM and SA-NM showed two distinct absorptions, one at 2900 cm-1 for CH3 and the other at 2800 cm-1 for CH2. The presence of the ester carbonyl group in ALN was indicated by three characteristic bands at 1700, 1200, and 1016 cm-1. The absorption band at 1016 cm-1 in ALN suggested the presence of amylose-fatty acid C-O functionality. In figure 1, please index the main peaks of FTIR.
5. Line 92, “Compared to previous results,… ” What does “previous results” stand for? A citation would be necessary herein.
6. In figure 4, what do a, b, c, d, cd… stand for? Authors should point out their meanings in the figure caption or in the main text.
7. In figure 3b, the TGA curve of control should be moved out and treated as a single panel. The TGA curves of LA and SA have not reach the plateau, these curves even have reciprocation at about 100 oC. Why?
Author Response
Reviewer # 2
In this study, authors isolated and characterized Amylose-Lipid Nanocomposite mate- 26 rial (ALN) from starch and lipid mixtures containing stearic, palmitic, and lauric acids. The ALN was formulated into nanocomposite hydrogels to evaluate its anthocyanin absorption capacity. Furthermore, this study evaluates the anthocyanin encapsulation and retention by ALN-based hydrogels, highlighting the impact of pH and physical crosslink networks on the uptake capacity. The results are interesting and authors are advised to address the following issues.
- The introduction section is lack of scientific background. Authors are advised to add an additional paper on hydrogel: https://doi.org/10.3390/jfb14060320, https://doi.org/10.1016/j.ijbiomac.2023.125754.
Response: corrections are made and addition of the provided literature is placed at line 65-67.
- Section 3.15: The experiments were carried out in triplicate and the resulting data was subjected to statistical analysis using one-way ANOVA. The means were compared using the Duncan multiple range test with a significance level of P < 0.05. However, no statistical analysis results can be found in the figures.
Response: Corrections are made where applicable such as in Figure 4, line 271, page 8.
- Authors please add a blank before and after the “±”. Please remove “title” from the title. XRD, SEM, TGA are common characterization methods, which are not advised to add in the keywords list.
Response: Correction effected. Thank you for your suggestions. The corrections have been made. The title word has been removed from the title, while XRD, TGA etc. have also been removed from key words. Page 1, line 2 and 42-43.
- The IR spectra of nanomaterial obtained from mixtures LA-NM and SA-NM showed two distinct absorptions, one at 2900 cm-1 for CH3 and the other at 2800 cm-1 for CH2. The presence of the ester carbonyl group in ALN was indicated by three characteristic bands at 1700, 1200, and 1016 cm-1. The absorption band at 1016 cm-1 in ALN suggested the presence of amylose-fatty acid C-O functionality. In figure 1, please index the main peaks of FTIR.
Response: Correction effected. Thank you for your suggestions. See page 3, figure 1.
- Line 92, “Compared to previous results” What does “previous results” stand for? A citation would be necessary herein.
Response: The reference has been cited at page 3, line 104.
- In figure 4, what do a, b, c, d, cd… stand for? Authors should point out their meanings in the figure caption or in the main text.
Response: Corrected as instructed at line 271, page 8.
- In figure 3b, the TGA curve of control should be moved out and treated as a single panel. The TGA curves of LA and SA have not reach the plateau, these curves even have reciprocation at about 100 oC. Why?
Response: The TGA of control has been separated Page 6, figure 3(c). At around 100 °C, water loss is observed in almost all the samples. The reciprocation arose due to different nature/behavior of samples at this temperature in loosing water molecules. Moreover, the data has been put in one frame for the sake of place and also to compare the thermal degradation nature of different samples. The TGA curves of LA and SA are stable beyond the tested temperature so these are not touching the plateau. A detail insight has already been provided in its discussion. Line 188-206, page 5 and 6.
Round 2
Reviewer 1 Report
Dear Authors,
I have received the manuscript “Green synthesis of low-glycemic Amylose-Lipid Nanocomposites by high-speed homogenization and formulation into nanocomposite hydrogel” to be reviewed for the Molecules (ISSN 1420-3049) Journal. Thank you for your first corrections. Below are specific comments regarding points that still need to be improved.
1. Include the explanation about the Green synthesis in the introduction (response 2 from the first reply). This is important to justify the use of the term and the innovation involved.
2. Regarding the X-ray patterns. I could not understand why the authors are unable to change the x-axis range in the graphs. Something should be done to improve the identification of the peaks that were discussed, for example using arrows.
3. About significant statistics. When you use a, b, c to indicate significance, you should provide which groups you are comparing (e.g. a = control vs treatment). These issues should be reviewed in Table 1 and Figure 4. The sentence "represents significant difference (p≤0.05)" is not enough. Furthermore, in figure 4 there is no "e" as was described in the caption.
4. I consider that the contextualization of rheological parameters can be improved. The authors added lines 331-335, but this is not sufficient. Please, discuss more about what are the expected rheological behaviour for hydrogels used in the food industry and if the results that you obtained are aligned with this expectation. The rheological part is too much descriptive, so the discussion should be improved.
Author Response
Dear Reviewer
We are grateful for the valuable comments/suggestions provided by you. The line-by-line suggestions and responses have been provided as attachmment. The corrections have also been made in the MS and highlighted as yellow.
